# Hydraulic Effect of Vegetation Zones in Open Channels: An Experimental Study of the Distribution of Turbulence

**Tomasz Tymiński \*** and **Krzysztof Wolski**

Institute of Environmental Engineering, Wrocław University of Environmental and Life Sciences,
pl. Grunwaldzki 24, 50-363 Wrocław, Poland; wolski.k.is@gmail.com
\* Correspondence: tomasz.tyminski@upwr.edu.pl

**Abstract:** The development of vegetation in riverbeds is an important part of river engineering, and an in-depth understanding of its hydraulic influence is greatly needed. Our research focuses primarily on common reed (*Phragmites australis*) in riverbeds. To date, little is known about the hydraulic impact of the *Phragmites australis* reed and both field and laboratory data are still very scarce. Consequently, the main goal of our study was to evaluate the effect of vegetation zones on the spatial distribution of turbulence. Based on laboratory measurements of local instantaneous velocities, the values of the turbulence intensity (degree) *Tu* were determined, and its spatial distribution was illustrated. Analysis of the results showed that the relatively dense clusters of plants (reeds) act as "openwork deflectors" of the current and very clearly shape its spatial distribution. This can also be observed in the case of the distribution of the turbulence parameter *Tu*. For example, in the case of the development of riparian vegetation in the form of quasi-triangular communities of common reed (*Phragmites australis*) located alternately, there is a channelization of the flow, but also spatial changes in its character that occur. This work only presents results for preliminary hydraulic tests for *Phragmites* reed. These experiments should also be continued for other species of flexible riparian vegetation such as wicker. In the laboratory, the hydraulic influence of only triangle-shaped vegetation zones has been studied. Therefore, there is also a need for further hydraulic studies on vegetation zones of shapes other than triangular, e.g., rectangular, as well as vegetation zones with irregular shapes The authors see the need for such research and have already planned its continuation. Research on the interactions between vegetation and the structure of water flow in the riverbed is a very important aspect of contemporary trends in river environment management. Conscious, planned, and model-tested locating (or removing) of vegetation in a stream allows for shaping hydraulic and morphological conditions, thus controlling the processes of erosion, transport, and accumulation of debris.

**Keywords:** rivers; riparian vegetation; hydraulic flow conditions; stream turbulence

## 1. Introduction

Vegetation development in the riverbed is an important part of river engineering and there is a need for a good understanding of its hydraulic impact. Research on the interactions between vegetation and the structure of water flow in the riverbed is a very important aspect of contemporary trends in river environment management. Model studies on species-specific, measured plant clusters with known morphological and biomechanical characteristics make it possible to determine the disturbances in the distribution of velocity and turbulence in the river channel, as presented in this article. Knowledge of the magnitude of these impacts allows for practical activities in river engineering: firstly, assessing whether vegetation in a given area of the river does not limit the minimum capacity and therefore requires its removal; and secondly, verification of the introduction of plants where they currently do not occur as a river restoration activity that builds biodiversity and hydraulic variability of the stream. Conscious, planned, and model-tested locating

(or removing) of vegetation in a stream allows for shaping hydraulic and morphological conditions, thus controlling the processes of erosion, transport, and accumulation of debris. In modern, eco-friendly river engineering, hydromorphological conditions in the riverbed can be regulated and shaped by conscious, planned plantings.

The occurrence of plant communities, such as shrubs of purple willow (*Salix purpurea* L.) or rushes of common reed (*Phragmites australis*), influences the flow conditions in the riverbed, and affects, in particular, the spatial distribution of flow velocity and stream turbulence [1–4]. Such plant hydraulic interaction is not easy to determine as it depends on many variable factors, e.g., plant species and mechanical characteristics [5–7] and their stage of development including the density of the plant zone [3,4,7,8]. This interaction also depends on the spatial configuration of the vegetation zone (quasi-regular vegetation development, e.g., in the form of a triangle, circle, rectangle, etc., or the opposite, very irregular) and on the location of the plant community (at riverside (Figure 1), or directly in the river current (Figure 2)). According to Kubrak et al. [9], the greatest impact on reducing the channel capacity and disturbing the flow field is that of the vegetation found along the banks of the main channel. For this reason, it is preferable when there are no shrubs and dense reed communities along the river shoreline. On the other hand, plants are a fundamental and indispensable part of the natural environment. It is also important to realize that shrubs and reeds become "sealed" by grasses and leaves carried by water, which, if they are dense enough, results in the exclusion of this part of the section from the flow and additional damming of water in the channel.

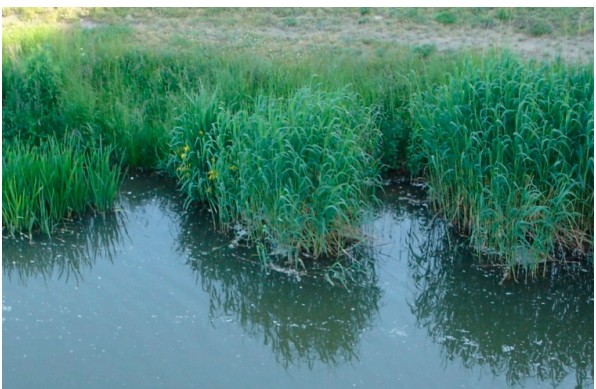

**Figure 1.** Quasi-triangle riverside vegetation zones.

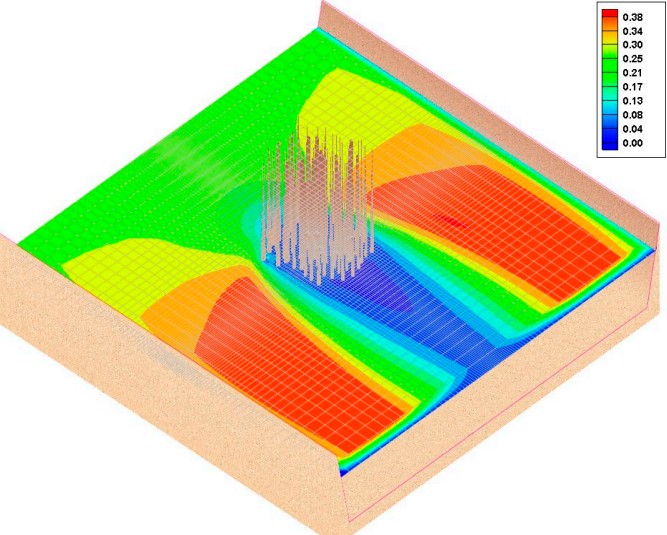

**Figure 2.** Local distribution of velocity around a vegetation zone located directly in the river current (an example simulation for the Sleza River).

In addition to the biomechanical characteristics of plants [5,7,10], it is equally important to have a good understanding of water flow characteristics. Among those occurring in nature, especially in rivers, the turbulent flows prevail, their dominant feature being the chaotic and irregular movement of fluid. The high intensity of transport processes is due to the complex movement of vortices of different scales and energies [11–14]. There are chaotic fluctuations of hydrodynamic parameters, e.g., velocity, the values of which change both in time and in space. According to the Reynolds hypothesis, the turbulent flow is a superposition of averaged and fluctuating flow [11,13–16]:

$$v = v_m + v' \ [\text{m/s}] \tag{1}$$

where $v'$ is the velocity fluctuation around the mean motion, and $v_m$ is the time averaged velocity:

$$v_m = \frac{1}{T} \cdot \int_{t-(T/2)}^{t+(T/2)} v dt \ [\text{m/s}] \tag{2}$$

Study of flow velocity and turbulence distribution in a vegetated channel is possible either on objects in nature (rivers) or by using either 2D or 3D-numerical simulation or physical models in a laboratory. A quantitative measure of the magnitude of flow velocity fluctuations is provided by the intensity (degree) of turbulence *Tu* (also known as the "*Tu-number*"), which can be determined separately for each direction of flow [15–18] or from the following formula [13,15]:

$$Tu = \frac{\sqrt{\frac{1}{3}\left(v_x'^2 + v_y'^2 + v_z'^2\right)}}{\sqrt{v_{m,x}^2 + v_{m,y}^2 + v_{m,z}^2}} \ [\text{-}] \tag{3}$$

where:

$v_x'$, $v_y'$, $v_z'$—velocity fluctuation components [m/s], $v_{m,x}$, $v_{m,y}$, $v_{m,z}$—average velocity components [m/s].

According to Koziol [19], the turbulence intensity *Tu* is the most commonly studied turbulence characteristic both in science and in practice, but equally important are the turbulent flow analyses related to energy transformations, including the so-called turbulent kinetic energy (*TKE* or *k*). The *TKE* parameter is characterized by the mean square velocity fluctuation and can be calculated from the following relationship [20,21]:

$$TKE = k = \frac{1}{2} \cdot \left(v_x'^2 + v_y'^2 + v_z'^2\right) \ [\text{J/kg}] \tag{4}$$

When flowing through the plant zone, the generated turbulent kinetic energy (*TKE*) is the result of tangential stresses and frictional forces and the vortex path behind the flowing plant stems [19,22–24]. By comparing Formulas (3) and (4), we finally obtain:

$$Tu = \frac{1}{v_m} \cdot \sqrt{\frac{2}{3}k} \ [\text{-}] \tag{5}$$

Apart from the degree of turbulence *Tu*, the so-called turbulence scale is also introduced to characterize the turbulent flow. It determines the size of the vortices that form in turbulent motion [25,26]. Although the analysis of the vorticity of flow in riverbeds is a complex problem of paramount importance [27,28], it is, however, not the main focus of this research. Instead, we focused primarily on the reeds in a riverbed.

It should be assumed that with the conscious and planned location of the reed zones and with appropriately choosing their shape, it is possible to shape the spatial distribution of turbulent kinetic energy (*TKE*) of the stream in the river. This is of great importance, for example, in the analysis of erosion, transport, and sedimentation processes of river

debris. Erosion processes of river material are to be expected in this concentrated main stream. On the other hand, in the zones with low intensity of turbulence, sedimentation and accumulation of river debris is possible.

In studying the literature, the authors' particular attention was drawn to several recent research papers that deal with the hydraulic effects of plants. The conclusions of these studies are interesting, e.g., Tang et al. (2021) find that the stem-near-field turbulence inside a dense vegetation zone could be similar to that around a single cylinder. In fact, the results referring to a single generic cylinder are representative only for such a case [29]. Additionally, Penna et al. (2020) studied turbulence characteristics for stiff riparian vegetation. Their results show that flow depth is important in this case. In the flow zone close to the bottom, there may be a combined effect of hydraulic action of the vegetation and frictional forces on the channel bottom. However, moving towards the free water surface, the flow conditions are strongly influenced by the vegetation. The analysis of spatial distribution of the TKE-parameter (Equation (4)) revealed high values below the water level and also in the near-bed flow zone in the stream wise direction. A strong lateral variation of the TKE from the flume centerline to the cylinders occurred in the intermediate zone [30].

Valyrakis et al. (2021) focused their study mainly on the role of riparian vegetation density in riverbed hydrodynamics. Based on an extensive experimental study, spatial distributions of flow velocities were determined and then roughness coefficients and tangential stresses were analyzed in terms of the density and location of riparian vegetation zones. According to Valyrakis et al.: (1) as stem density increases, mean flow velocity in the main channel increases while mean flow at the riverbank decreases; and (2) the arrangement of riparian vegetation can be as important as that of the density in modifying the mean flow field of the main channel for low riparian densities [31].

In a mountain riverbed with plants and a gravelly bottom, Wos and Ksiazek (2022) studied the spatial distributions of velocity, turbulence intensity, turbulent kinetic energy TKE, shear stress and Froude number. Although there was a wide dispersion of the turbulence variable distributions, a standard tendency of decreased mean velocity and increased turbulence towards the bottom was observed [32].

Soltani et al. (2020), on the other hand, conducted their study of turbulent flow characteristics on the Plusjan River (Iran) with a very rough, gravelly riverbed. Their results show that hydraulic interactions between riparian vegetation, unsteady flows, and the roughness of the riverbed lead to highly irregular spatial distributions of turbulent velocity and intensity [33].

Afzalimehr et al. (2019) also studied distributions of flow velocity and turbulence intensity. Based on field measurements, they compared such distributions for riverine gravel bedforms with and without vegetation. They found that a change in bed roughness from gravel to vegetated bedforms results in a different shape of velocity distributions, increases turbulence anisotropy, generates strong secondary currents, and influences the location of erosion-prone zones in the riverbed [34].

Still, little is known about this aspect of the hydraulic impact of vegetation (*Phragmites australis*), and there is still little field or laboratory data available. Therefore, the main aim of our research was to gain more insight into the phenomena related to the distribution of turbulence and their influence on flow conditions in a small lowland river. We were particularly interested in evaluating the effect of vegetation zones on the spatial distribution of the "*Tu-number*". It should be emphasized that the presented laboratory experiments were carried out without using plant substitutes (rigid and smooth plastic cylinders are commonly used), but with natural reed vegetation of the *Phragmites* species, which is flexible and rough with branches and leaves. For this reason, the results of the presented laboratory research are closer to nature and to the actual flow conditions in the river.

## 2. Materials and Methods

### 2.1. Modelling Research

The laboratory model research was conducted for the Sleza River (Poland), which is a small lowland river with vegetational build-up containing mainly flexible reeds (Figure 1). The modelled short section of the Sleza River is located in the geographical location 16°58′36″ east longitude and 51°04′53″ north latitude. The physical model of the river was built (at a scale of 1:4) in a trapezoid flume measuring 15 m in length. The side slope was 1:1 and the bottom width $-0.9$ m. The bottom slope was constant over the entire flume length: $J = 12‰$. Our research focused mainly on qualitative analysis of the hydraulic effects of flexible *Phragmites* reed. The drag resistance of this type of plant is dominant compared to the frictional resistance of the bottom of a lowland riverbed [1–3,6,10,18]. The Manning–Strickler roughness coefficient for the entire length of the flume bottom was $k_{St} = 80$ [16]. The main flume with plants measured 3 m in length. The plant zones consisted of configurations of triangular ($0.6 \times 0.6 \times 0.6$ m) very flat flowerpots, in which plant stems were placed (Figures 3 and 4). Common reed (*Phragmites australis*) with a stem diameter of about 4 mm was used for the experiments. The density of the vegetation zones was 578 plants/m², with the stem spacing of $a_x = a_y = 0.05$ m. This species of reed (*Phragmites australis*) is common throughout the entire world and can withstand long periods of immersion [5,35,36].

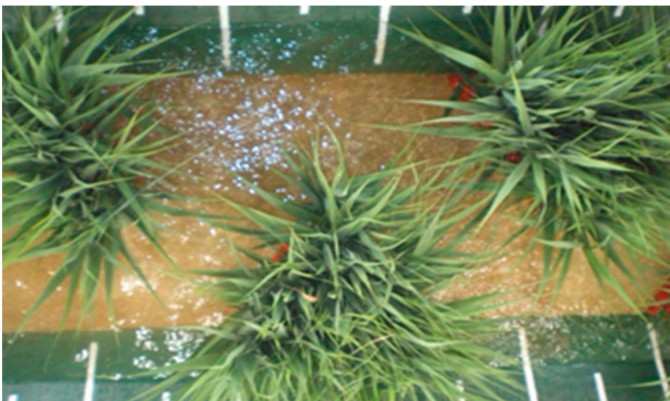

**Figure 3.** Laboratory research flume with vegetational build-up (*Phragmites australis*).

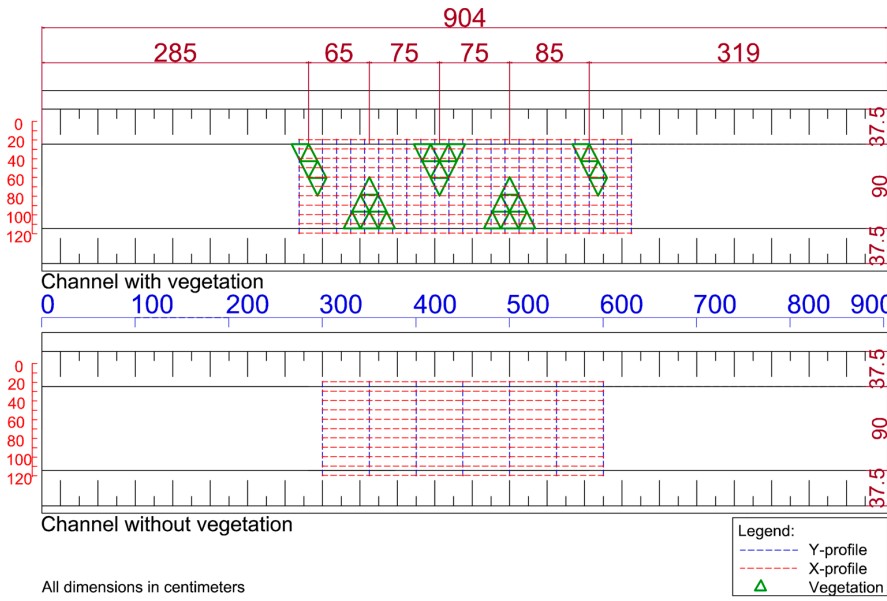

**Figure 4.** Diagram of the arrangement of the plant zones and the measurement site in the laboratory flume.

### 2.2. Methods and Scope of Research

The hydraulic experiments carried out in a lab consisted of the measurements of flow depth, local velocity and stream distribution for a given flow rate $Q$. A picture of the velocity field was obtained from the measurements taken using an electromagnetic multidirectional velocity meter PEMS (Figure 5) at a total of 360 measurement points. The measurements were taken at the nodes of a $0.10 \times 0.15$ m mesh and at the characteristic points of flow disturbance. The PEMS type electromagnetic velocity meter (Figure 5) is equipped with an E30 measuring probe. It is a disk-shaped probe with dimensions of $33 \times 11$ mm, mounted on a rod. This meter enables the measurement of the longitudinal component of velocity $\pm v_x$ and the transverse component $\pm v_y$, the value of the resultant velocity $v_e$ and the angle of its deviation $\alpha$ from the direction of the positive semi-axis $v_x$. The minimum sampling time of the meter is 0.1 s, and the velocity measurement accuracy is $\pm 0.01$ m/s. In our laboratory tests, the range of hydraulic parameters was as follows: (1) flow rate $Q = 30$–$50$ dm$^3$/s, (2) average instantaneous velocities $v = 0$–$1$ m/s, and (3) flow depth $H = 0.15$–$0.28$ m.

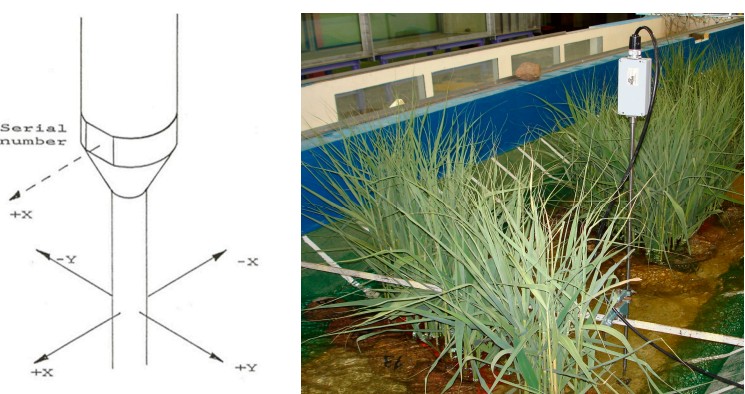

**Figure 5.** Measuring instruments (the PEMS).

Based on the laboratory measurements of local instantaneous velocities, values of the degree of turbulence $Tu$ were determined and its spatial distribution was illustrated. The hydraulic influence of vegetation zones in the channel on the spatial distribution of flow velocity and the degree of turbulence were thoroughly analyzed. All data analyzed in our study were developed and converted according to the scale (1:4) based on the hydrodynamic criterion for Froude similarity [11,14].

### 3. Results and Discussion

Laboratory measurements of the velocity field allowed us to determine the Formula (3) for the degree of turbulence $Tu$. For the purpose of analysis of flow conditions, graphic illustrations of the flow velocity and turbulence distribution for all the tested configurations of the vegetation zones have been made. The hydraulic analysis was carried out for two main study options: (1) laboratory flume with vegetation, and (2) laboratory flume without vegetation. An example of the results of our laboratory experiments for a specific flow $q_{LAB} = 0.033$ m$^3$/(s·m) and flow depth $h = 0.25$ m is shown below in Figures 6–10. This paper presents the results of preliminary laboratory tests for only one measurement series. Equation (3) assumes $v_z = 0$ due to the limitations of the PEMS apparatus. Our study will continue for greater flow depths and will use a three-way ADV probe.

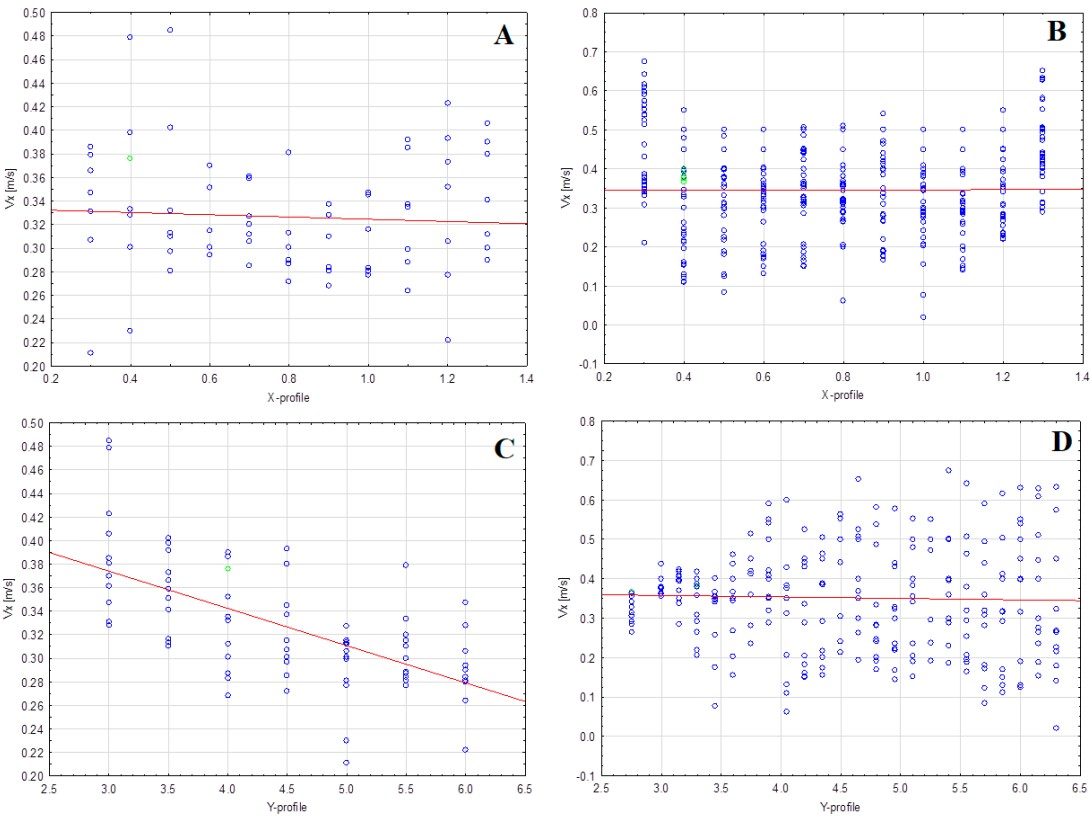

**Figure 6.** (**A**) Distribution of the flow velocity $v_x$ in profiles X (lengthwise) in a channel without vegetation, (**B**) distribution of the flow velocity $v_x$ in profiles X in a channel with vegetation, (**C**) distribution of the flow velocity $v_x$ in profiles Y (crosswise) in a channel without vegetation, (**D**) distribution of the flow velocity $v_x$ in profiles Y in a channel with vegetation.

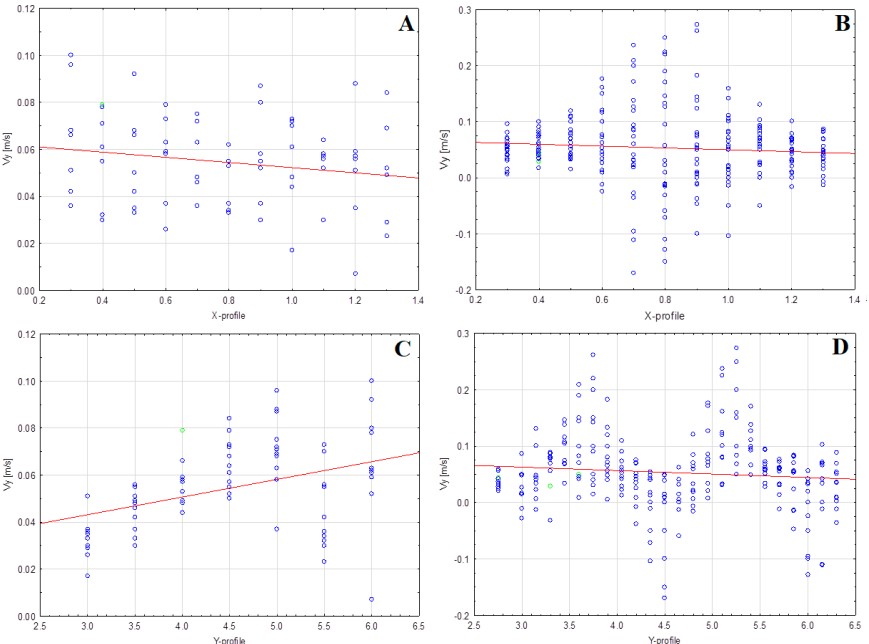

**Figure 7.** (**A**) Distribution of the flow velocity $v_y$ in profiles X in a channel without vegetation, (**B**) distribution of the flow velocity $v_y$ in profiles X in a channel with vegetation, (**C**) distribution of the flow velocity $v_y$ in profiles Y in a channel without vegetation, (**D**) distribution of the flow velocity $v_y$ in profiles Y in a channel with vegetation.

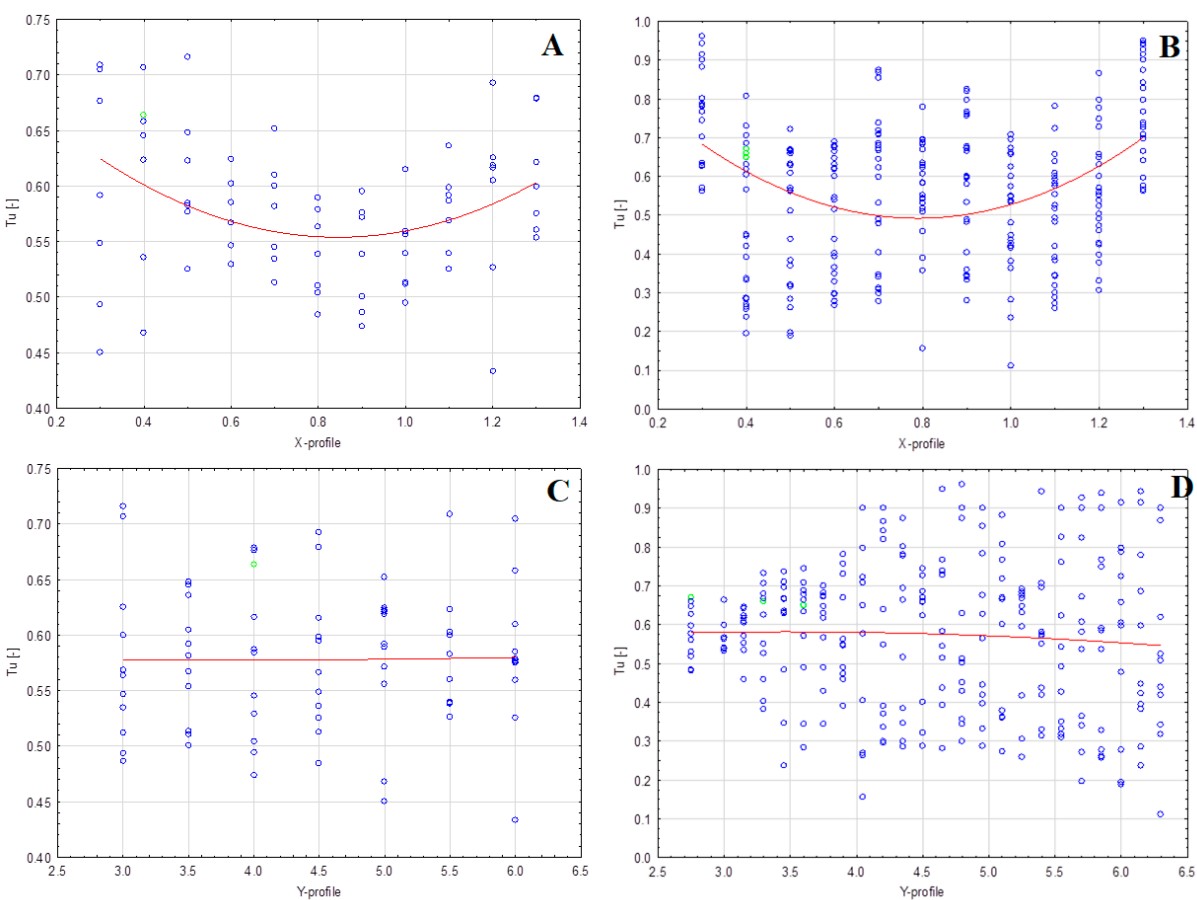

**Figure 8.** (**A**) Distribution of the degree of turbulence *Tu* in profiles X in a channel without vegetation, (**B**) distribution of the degree of turbulence *Tu* in profiles X in a channel with vegetation, (**C**) distribution of the degree of turbulence *Tu* in profiles Y in a channel without vegetation, (**D**) distribution of the degree of turbulence *Tu* in profiles Y in a channel with vegetation.

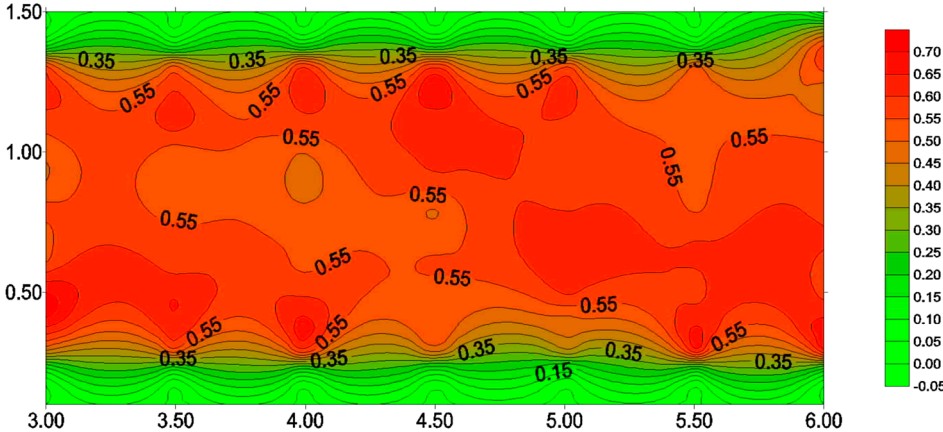

**Figure 9.** Spatial distribution of the intensity of turbulence *Tu* in the laboratory flume without vegetation {where: *Tu(mean)* = 0.45, *Tu(min)* = 0.28, *Tu(max)* = 0.58}.

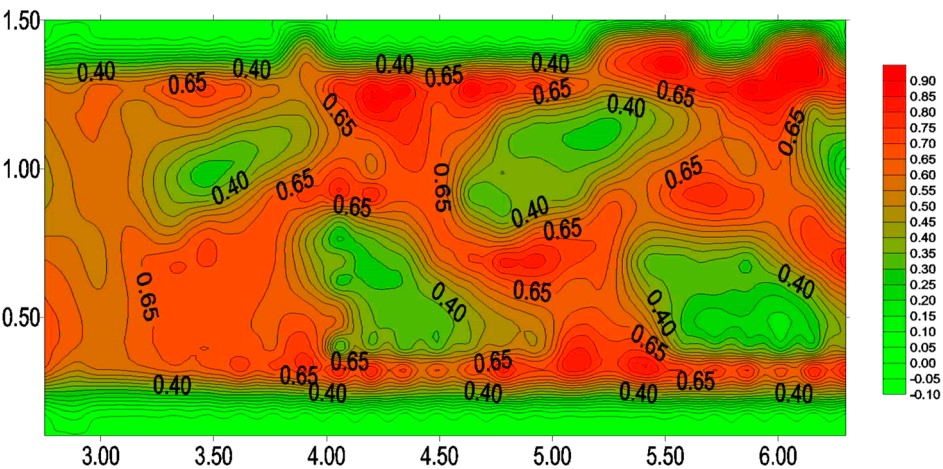

**Figure 10.** An example of spatial distribution of the intensity of turbulence *Tu* by *q* = 0.033 m³/(s·m) in the laboratory flume with vegetation {where: *Tu(mean)* = 0.55, *Tu(min)* = 0.24, *Tu(max)* = 0.80}.

### 3.1. Spatial Distribution of the $v_x$ Velocity (the Longitudinal Component of Flow Velocity)

In the analyzed flow field, the average $v_x$ values for both variants being studied (plants or no plants) are at similar levels of approximately 0.33–0.35 m/s. However, the distribution of the velocity component $v_x$ along the channel relative to the X-profiles (lengthwise) differs significantly for these two cases. The velocity distributions in Figure 6 show a slight rightward deviation of the values from the flow axis. This, however, occurs for all conditions (test variants) and is probably due to the laboratory flow instability and insufficient alignment in flow in the flume from the pumping station. Importantly though, this does not affect, in any way, the conclusions of our study.

For the vegetation-free channel (Figure 6A), the measured flow velocities are in the range of 0.21–0.49 m/s (with up to 90% of our observations relating to velocities between 0.27 and 0.40 m/s). Flow velocities in the vegetation-free channel show only a slight variation with respect to the X-profile (lengthwise).

In the variant with vegetation (Figure 6B), the distribution of flow velocity $v_x$ along the channel is much more complex. The range of $v_x$ is significantly wider both over the entire analyzed flow area (from 0.04 to 0.67 m/s) and for individual X-profiles, which is due to the alternating arrangement of vegetation communities.

In the distribution of the velocity component $v_x$, five zones are clearly distinguishable. At the riverbank (in our laboratory model), where there are very few plant stems and the plant community density is low, there are zones of significantly increased flow velocity. According to Liu et al. [37], this is due to an increased "flow agitation" effect from the individual vegetation elements when located in a sparse arrangement. This mechanism is largely silenced by the "flow blockage" effect which increases with the increase in vegetation density [31,37]. Next (Figure 6B), closer to the center of the channel, there is a zone with a wide range of changes in $v_x$, which results from the alternating location of plant communities and zones of free flow in the channel. Lastly, in the central part of the cross-section, there is a zone where the values of $v_x$ are higher, although not as high as near the channel banks.

In the distribution of velocity $v_x$ relative to the Y-profiles (crosswise), the differences between the channel variants with and without vegetation concern two main features: (1) the magnitude of the velocity component $v_x$ recorded in successive Y cross-sections, and (2) the variability of $v_x$.

For the variant without vegetation (Figure 6C), there is a very clear change in the overall average velocity $v_x$ in successive Y-sections, while there is little variation in $v_x$ at individual cross-sections. In the analyzed area, the velocity $v_x$ along the riverbed clearly decreases (cf. the red, sloping, straight line in Figure 6C).

For flow conditions in a flume with vegetation (Figure 6D), the average velocities $v_x$ basically do not change at all in successive Y-profiles. However, the variability of $v_x$ clearly increases with the size of the vegetated flume development zone (along the direction of flow). Undoubtedly, this high variability of the velocity component $v_x$ is due to the alternating distribution of plants in the channel, the significantly higher flow disturbances, and the higher "flow intensity". Here, the hydraulic effect of plants on flow conditions is evident, and our laboratory observations in this case corroborate scientific reports from other research teams [19,23,35,38–43].

### 3.2. Spatial Distribution of the $v_y$ Velocity (the Lateral Component of Flow Velocity)

It is especially the instantaneous transverse velocity $v_y$ that largely characterizes the turbulent nature of the flow stream. The values of $v_y$ measured in the laboratory flume show very significant differences between the variants with and without vegetation. For the flow in the channel without vegetation (Figure 7A), the transverse velocities $v_y$ analyzed in successive X-profiles (lengthwise) are relatively insignificant (about 0.05 m/s on average) throughout the channel. In the analyzed flow field, the positive values of the $v_y$ component indicate, throughout the entire channel, a slight deviation of the flow in one direction, as mentioned earlier when characterizing the $v_y$ velocity.

When analyzing the distribution of the velocity component $v_y$ in the Y-profiles (crosswise), it is noticeable that the values of $v_y$ increase slightly with the length of the flume (Figure 7C). Moreover, the values of average crosswise velocity $v_y$ in the flume with plant zones are at a very similar level as in the variant without plants. This shows that the influence of flow instability in the flume can be ignored in the comparative analysis of the two variants (plants or no plants) [3,19,35]. However, the value and distribution of the transverse velocity $v_y$ in the studied variant of the channel with vegetation is clearly influenced by the location of alternating zones of bank vegetation.

In the distribution of velocity $v_y$ relative to the X-profiles (lengthwise), the greatest variation in $v_y$ occurs in the central part of the channel (from −0.18 to 0.27 m/s). In this area, the direction of flow changes frequently, as water meanders between the successive zones of plants.

As one approaches the banks of the channel, the range of variability of $v_y$ narrows and $v_y$ nears its mean value. On the other hand, in the analysis against Y-profiles (Figure 7D), the fluctuation of the transverse velocity $v_y$ along the length of the channel caused by the hydraulic influence of alternating vegetation zones is most clearly visible (Figures 3 and 4). This hydraulic phenomenon has also been observed by several other research teams [44–48].

### 3.3. Intensity of Turbulence

In the analysis of the distribution of the degree of turbulence *Tu* relative to successive X-profiles (lengthwise), it is apparent that there is a very clear difference in the values of the *Tu*-parameter determined for flow in a channel with plant zones compared to flow conditions in a channel without plants (Figure 8A,B). In this case, the hydraulic impact of plants is particularly significant, which is also confirmed by literature reports [15,19,44,48–51].

In the distribution of the *Tu*-parameter in the laboratory channel without plants (Figure 8A), the maximum *Tu* turbulence values occur near the banks. As the distance from the banks of the channel toward the main current increases, turbulence decreases, and the water flow stabilizes.

In the variant with plants in the channel (Figure 8B), the trend described above is somewhat similar. However, the minimum turbulence values of *Tu* do not occur in the center of the channel cross-section, as there is an increase in longitudinal velocity $v_x$ and it is a zone of increased stream turbulence. Other researchers have also noted this hydraulic effect of plants [19,31–37,51,52].

In the analysis of the distribution of the *Tu* parameter against Y-profiles (crosswise) for the variant without plants (Figure 8C), there are no statistically significant changes

along the length of the flume. On the other hand, the changes in the degree of turbulence *Tu* in the variant with vegetation in the laboratory flume (Figure 8D) are very interesting and visible. An increase in the variability of *Tu* turbulence already occurs before the first vegetation obstacle (Figure 4), after which this variability gradually increases. The average value of *Tu* turbulence in the cross-section of the channel reaches its maximum just at the first vegetation obstacle. After that, the average value of *Tu* turbulence lightly decreases, gradually along the length of the channel, and with the appearance of subsequent alternating vegetation zones. The turbulence-dampening and flow-calming effect of plant zones in a channel has been reported in several scientific studies [3,15,35,53–55].

Analysis of the results also showed that relatively dense clusters of plants (reed) act as "openwork deflectors" of the stream and shape its spatial distribution very clearly. This can also be observed with the distribution of the turbulence parameter *Tu*. For example, for riverside vegetation development in the shape of quasi-triangular reed communities (*Phragmites australis*) located alternately (Figures 3 and 4), there is channelization of the flow, but also a spatial change in its flow character (Figures 9 and 10).

Compared to the flume without plants (Figure 9), the mean value of *Tu* increased by ca. 20%. However, immediately upstream of the plant zone, *Tu* increased by up to about 40% (*Tu* = 0.65), and downstream of the plant zone, this parameter decreased by about 11% (*Tu* = 0.40). The highest turbulence was measured in the main current, especially in the corridors between vegetation zones {*Tu(max)* = 0.80}. The values of *Tu* = 0.24–0.80 obtained by the authors are comparable with literature data. For example, according to Mazurczyk [15], the parameter *Tu* increases with, among other factors, plant density in the riverbed and takes on values in the range of *Tu* = 0.05–0.90. The values of *Tu* obtained by Kałuża [3] are in the range of 0.15–0.75 for plants in his lab flume arranged in a checkerboard pattern. Similar to Kałuża [3], in the spaces between plants on the upstream side of water, we also observed zones of high turbulence muted in the direction of flow.

## 4. Final Conclusions

To date, many researchers have dealt with the problem of hydraulic flow conditions in rivers. Some of them have also studied the spatial distribution of turbulence [15,17,19,25,28–34,38,51–54]. However, those studies focused on a completely different factor that affects the distribution of turbulence. In those cases, they mainly studied the effect of a very rough, gravelly riverbed on the distribution of turbulence and turbulent kinetic energy (*TKE*). Even when there were plants in the river (often common river hydrophytes rather than reeds), their hydraulic influence was not dominant as in the case of gravel-bed rivers [17,28,32–34,52]. There are known laboratory studies where the influence of plant zones on stream turbulence and flow conditions was also tested. However, in these experiments, only plant substitutes were used. These were plastic, rigid cylinders [15,19,25,29–31], with geometric and biomechanical characteristics that are very different from those of natural plants. In particular, reeds are flexible, and a stream of water can bend them. These mechanical characteristics of plants are very important in terms of their hydraulic effects [5,7] and it is imperative to take them into account in studies and various hydraulic calculations [6].

Therefore, it should be emphasized that our laboratory experiments were carried out without using plant substitutes, but with natural reed vegetation of the *Phragmites* species. In a water laboratory, such tests are not easy, but the reliability of the results obtained for natural, flexible plants is better than the results for rigid and smooth plastic cylinders. By using *Phragmites* reed in our study, the flexibility of the stems was taken into account, as well as the roughness and spatial structure of the plants (shape, branches, leaves etc.). Additionally contributing to the general state of knowledge is the fact that these hydraulic experiments involved a specific plant species called *Phragmites australis*. There are no reports in the scientific literature of similar hydraulic laboratory tests for just this species of reed, which is found on banks and in riverbeds and is very common in many countries.

Our laboratory tests were aimed at recognizing the problem of hydraulic interaction of *Phragmites australis*. This work will be continued. Our research has confirmed that

dense communities of this common reed species act as openwork deflectors of the stream and shape its spatial distribution in the riverbed. Their influence is visible, among other things, in the distribution of velocity and turbulence. The zones of vegetation (deflectors) channelize and differentiate flow in the channel creating distinct current and calm zones (with reduced stream velocity and turbulence). Alternating dense zones of *Phragmites australis* gives the stream a clearly meandering character. Our analysis also reveals that by deliberate, planned planting or removal of plants in the riverbed, it is possible to influence the spatial structure of the stream (parameters "*v*" and "*Tu*"). Thus, it is possible to control the processes of erosion and accumulation of river material. This aspect, however, should be verified through field research. Moreover, the shape and placement of the *Phragmites australis* zone has an impact on stream channelization. In the first stage of our project, we only studied triangular reed zones; however, in the next stage we also plan to study stream deflectors of other shapes. So far, it has been shown that triangular zones of *Phragmites australis* (Figures 1 and 3) located at the riverbank affect the flow conditions significantly. One of the decisive factors in this impact is the density of the *Phragmites australis* reed community. Where the reed density was lower, higher flow velocities and values of the turbulence parameter "*Tu*" were measured. Where the reed zones were very dense or even poorly permeable, a more significant effect of the plants as stream deflectors was observed.

As mentioned in Section 1, the riparian plants' interaction with flow conditions in a river depends, among other factors, on the stage of development (density) and the shape of the plant zone, e.g., on the plan as a quasi-rectangle or a quasi-triangle (Figure 1). This work only presents results for preliminary hydraulic tests for *Phragmites* reed. These experiments should also be continued for other species of flexible riparian vegetation such as wicker. In the laboratory, the hydraulic influence of only triangle-shaped vegetation zones has been studied (Figures 3 and 4). Therefore, there is also a need for further hydraulic studies on vegetation zones of shapes other than triangular, e.g., rectangular as well as vegetation zones with irregular shapes.

The authors are also aware that their model test is carried out under ideal indoor environmental conditions. In the actual growing environment of the flexible vegetation, there is the influence of weather (such as wind). Similar observations and the so-called Honami effect (stem wobbling) are also reported by Tsujimoto and Kitamura [24]. Field studies of this problem on a river with flexible, riparian vegetation will certainly be of great interest. The authors see the need for such studies and plan to compare the hydraulic effects of *Phragmites* reed in the laboratory and in nature (river) and also verify the test results of the ideal model with those of the actual engineering environment.

**Author Contributions:** Conceptualization, T.T. and K.W.; methodology, T.T.; validation, T.T. and K.W.; formal analysis, T.T. and K.W.; investigation, T.T.; resources, T.T. and K.W.; writing—original draft preparation, T.T. and K.W.; writing—review and editing, T.T. and K.W.; visualization, T.T. and K.W. All authors have read and agreed to the published version of the manuscript.

**Funding:** This research received no external funding.

**Institutional Review Board Statement:** Not applicable.

**Informed Consent Statement:** Not applicable.

**Data Availability Statement:** Data is contained within the article.

**Conflicts of Interest:** The authors declare no conflicts of interest.

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
