# Peer review of "Hydraulic Effect of Vegetation Zones in Open Channels: An Experimental Study of the Distribution of Turbulence"

_sustainability, doi:10.3390/su16010337_

Round 1

Reviewer 1 Report

Comments and Suggestions for Authors

I have read you paper carefully and I have found that your paper is well desgined and scope of Sustainability Journal. However, I have some minor comments before accepted this paper.

1- Add more lateste reference to improve novelty of paper.

2- Please mention about other experimental parameters; and uncertainty error. 

3- I think the material and methods section have enough information for reader. So, It is fine.

4- Discussion is superficial. Please add your more obtained data.

5- Conclusion remark is missed. Please add your future recommandation.

6- If possible, please use more clear figures.

Comments on the Quality of English Language

 Minor editing of English language required

Author Response

View attachment

Reviewer 2 Report

Comments and Suggestions for Authors

This paper investigating the problem of hydraulic interaction of Phragmites australis plants. The conclusion of this study has some reference value for the Hydraulic Effect of Vegetation Zones in Open Channel. However, there are some major revisions are needed before the paper can be considered for publication, as follows:

1.       At the end of the introduction, the significance and value of this research can be briefly stated.

2 . The physical model of a short section of a small lowland river with vegetational build-up was built (at a scale of 1:4). What is the basis for the 1:4 scale?

3. The plant area consists of a triangular (0.6 x 0.6 x 0.6 m) very flat pot with the plant stem (Figure 3). What is the basis for a triangle of 0.6 x 0.6 x 0.6?

4.What is the basis for the vegetation area density of 578 plants /m2?

5. Figure 3 is too vague and should be replaced

6.The model test is carried out under ideal indoor environmental conditions. In the actual growing environment of the reed, there is the influence of weather (such as wind, rain, etc.). Should the test result take these factors into account? How different are the test results of the ideal model from those of the actual engineering environment?

7.The mark on the picture or the size of the number on the scale is inconsistent, please correct.

8.The conclusion of the article should be simple and clear, making it easier for the reader to understand

9. All the cited references are relevant to the research, but I don’t think the introduction has provide sufficient background and include all relevant references. Please refer to and cite the following references to improve it:

Effects of seasonal precipitation change on soil respiration processes in a seasonally dry tropical forest

Assessment of underground soil loss via the tapering grikes on limestone hillslopes

Study on Regional Differences and Convergence of Green Development Efficiency of the Chemical Industry in the Yangtze River Economic Belt Based on Grey Water Footprint

Modeling of rheological fracture behavior of rock cracks subjected to hydraulic pressure and far field stresses

Mechanical behavior of sandstone during post-peak cyclic loading and unloading under hydromechanical coupling.

The effect of Eulaliopsis binata on the physi-chemical properties, microbial biomass, and enzymatic activities in Cd-Pb polluted soil

Lugeon Test and Grouting Application Research Based on RQD of Grouting Sections.

10.In this study, only triangular shaped reeds were studied, why not study the distribution of rectangular, square, hexagonal or other shapes of reeds and compare them?

11. The research results of this paper should be compared and discussed with the existing research results to make the article more interesting.

12. The authors need to put the limitations of this study.

Comments on the Quality of English Language

This paper investigating the problem of hydraulic interaction of Phragmites australis plants. The conclusion of this study has some reference value for the Hydraulic Effect of Vegetation Zones in Open Channel. However, there are some major revisions are needed before the paper can be considered for publication, as follows:

1.       At the end of the introduction, the significance and value of this research can be briefly stated.

2 . The physical model of a short section of a small lowland river with vegetational build-up was built (at a scale of 1:4). What is the basis for the 1:4 scale?

3. The plant area consists of a triangular (0.6 x 0.6 x 0.6 m) very flat pot with the plant stem (Figure 3). What is the basis for a triangle of 0.6 x 0.6 x 0.6?

4.What is the basis for the vegetation area density of 578 plants /m2?

5. Figure 3 is too vague and should be replaced

6.The model test is carried out under ideal indoor environmental conditions. In the actual growing environment of the reed, there is the influence of weather (such as wind, rain, etc.). Should the test result take these factors into account? How different are the test results of the ideal model from those of the actual engineering environment?

7.The mark on the picture or the size of the number on the scale is inconsistent, please correct.

8.The conclusion of the article should be simple and clear, making it easier for the reader to understand

9. All the cited references are relevant to the research, but I don’t think the introduction has provide sufficient background and include all relevant references. Please refer to and cite the following references to improve it:

Effects of seasonal precipitation change on soil respiration processes in a seasonally dry tropical forest

Assessment of underground soil loss via the tapering grikes on limestone hillslopes

Study on Regional Differences and Convergence of Green Development Efficiency of the Chemical Industry in the Yangtze River Economic Belt Based on Grey Water Footprint

Modeling of rheological fracture behavior of rock cracks subjected to hydraulic pressure and far field stresses

Mechanical behavior of sandstone during post-peak cyclic loading and unloading under hydromechanical coupling.

The effect of Eulaliopsis binata on the physi-chemical properties, microbial biomass, and enzymatic activities in Cd-Pb polluted soil

Lugeon Test and Grouting Application Research Based on RQD of Grouting Sections.

10.In this study, only triangular shaped reeds were studied, why not study the distribution of rectangular, square, hexagonal or other shapes of reeds and compare them?

11. The research results of this paper should be compared and discussed with the existing research results to make the article more interesting.

12. The authors need to put the limitations of this study.

Author Response

View attachment

Reviewer 3 Report

Comments and Suggestions for Authors

Overall, this paper presents an experimental study on the hydraulic effects of vegetation zones, specifically common reed (Phragmites australis), in open channels. The research aims to understand how vegetation influences flow conditions, turbulence distribution, and spatial characteristics in a laboratory setting. The research addresses an important topic in the field of river engineering, and I commend your efforts in conducting this study.However, after a thorough review of your paper, I would like to provide feedback on areas where I believe substantial revisions are needed for the paper to reach its full potential and meet the standards expected for publication in sustainability Journal. I would like to emphasize that my feedback is intended to be constructive and aimed at enhancing the quality and impact of your work.

The abstract should be more informative and provide a concise summary of the paper's content, including the research methods, key findings, and their implications. In its current form, the abstract lacks sufficient detail.

The introduction section requires improvement in terms of providing a clear and concise overview of the research objectives and their significance. Establishing context is crucial for readers who may not be familiar with the subject matter.

The paper utilizes technical terminology and equations without adequate explanations. This may pose challenges for readers who do not possess an advanced understanding of the field. Please ensure that technical concepts are explained clearly.

The methodology section lacks essential details about the experimental setup, equipment specifications, and data processing procedures. Ensure that you provide all requisite details about the experimental setup, to facilitate the reproducibility of your study. Additionally, consider making your data available for fellow researchers to scrutinize.

The paper could benefit from a clearer explanation of the practical significance of the research and how the findings can be applied to real-world river engineering projects.

The quality of Figure 3 is poor. Figures 9 and 10 do not include units, making it difficult for readers to understand the context of the data being presented. It is crucial to include units for all measured quantities to maintain accuracy and clarity.

The paper presents a significant amount of data regarding turbulence but does not offer substantial interpretation or contextualization of these findings. It is imperative to elucidate the implications of the turbulence data in the context of your research objectives.

The technical challenges encountered during your experimental study are not adequately addressed. Discussing these challenges and how they were overcome, or their potential impact on the results, can enhance the scientific rigor of your paper.

The results and discussion section lacks coherence and fails to provide meaningful interpretations of the data presented. The paper should establish a clear narrative that links the results to the research objectives and discusses their implications. Furthermore, the discussion section should incorporate comparisons with existing literature and acknowledge any limitations of the study.

Comments on the Quality of English Language

These are just a few examples of the grammatical issues present in the paper. A careful proofreading and editing process is needed to address these concerns and improve the overall clarity and readability of the text.

Author Response

View attachment

Reviewer 4 Report

Comments and Suggestions for Authors

This manuscript focuses on the vegetation development in the riverbed which is an important part of river engineering and its hydraulic influence. This paper is very interesting and can be published in a Q1 journal like Sustainability. However, a few corrections and modifications have to be done to improve the quality of this paper:

 1.   The authors must explicitly define and explain the study's goal and objective in the Abstract of the revised edition. The new version's abstract must include an explanation of the reason behind conducting this research.

2. The resolution of Figure 3 is not good. It needs to be improved.

3. No need for Figure 4. It doesn't provide any additional information. Just keep Figure 5.

4. The conclusion has to be improved. The conclusion should not summarize the result of the paper only, it should also show the contribution of the current study compared to the scientific knowledge in the same field of research.

5. The English language needs to be checked and revised throughout the manuscript.

Comments on the Quality of English Language

The English language needs to be checked and revised throughout the manuscript.

Author Response

View attachment

Reviewer 5 Report

Comments and Suggestions for Authors

I've read the manuscript (sustainability-2629858) in detail and my comments are as follows: 

The background information and the literature review in the manuscript is quite weak, so I strongly recommend rewriting the introduction section. 

In section 2.2, methods and the scope of the research should be expanded with details. What is the purpose of using the division sign for v, H, and Q in this section? 

What are the terms in the curly bracket and paranthesis in the following sentence? "An example of the results of research for specific flow q = 1.05 m3/(s·m) {qLAB = 0.033 m3/(s·m)} and flow depth h = 0.45 m (h = 0.4·H)"

Please explain that what novelty is presented to the current literature by this work because there are several similar studies in the literature. The outcomes given in the final conclusions section are very general and can be found in several studies which examines the hydraulic impact of vegetation development. A few examples are listed below.  

Similar studies:

doi: 10.3389/fpls.2022.976646

DOI:10.5923/j.ijhe.20200901.03

https://doi.org/10.1007/s42241-019-0053-x

doi:10.1017/jfm.2022.598

https://doi.org/10.3390/w12092401

https://doi.org/10.1007/s10652-021-09791-9

Author Response

View attachment

Round 2

Reviewer 2 Report

Comments and Suggestions for Authors

The paper can be considered to accept

Comments on the Quality of English Language

The paper can be considered to accept

Reviewer 3 Report

Comments and Suggestions for Authors

Thank the author for addressing all comments. The paper, in its present form, is acceptable.

Reviewer 4 Report

Comments and Suggestions for Authors

All comments have been addressed. No further comments.

Reviewer 5 Report

Comments and Suggestions for Authors

I've read the revised manuscript. Although substantial alterations have been made in the text, the flow in the manuscript is still problematic. The conclusion part is very long and it looks like an introduction rather than a conclusion. Especially, the paragraph given below does not fit into this section. I think, the discussion presented in the conclusion section should've been presented in section 1. 

"Vegetation development in the riverbed is an important part of river engineering and there is a need for a good understanding of its hydraulic impact. Research on the interactions between vegetation and the structure of water flow in the river bed is a very important aspect of contemporary trends in river environment management. Model studies on species-specific, measured plant clusters with known morphological and biomechanical characteristics make it possible to determine the disturbances in the distribution of velocity and turbulence in the river channel, presented in the article. Knowledge of the magnitude of these impacts allows for practical activities in river engineering: firstly, assessing whether vegetation in a given area of the river does not limit the minimum capacity and therefore requires its removal, secondly, verification of the introduction of plants where they currently do not occur, as a river restoration activity that builds biodiversity and hydraulic variability of the stream. Conscious, planned, model-tested locating (or re-moving) vegetation in a stream allows for shaping hydraulic and morphological conditions, thus controlling the processes of erosion, transport and accumulation of debris. In modern, eco-friendly river engineering, hydromorphological conditions in the riverbed can be regulated and shaped by conscious, planned plantings."  

*** In section 4:

"Equation (5) shows the relationship TKE = f(Tu). This means that a change in the spatial distribution of turbulence also causes a change in the spatial distribution of turbulent kinetic energy (TKE)."

- This information/outcome can be found in several articles in the literature. Why is this information reported in the conclusions? 

"Our study showed the effect of the Phragmites australis zone on the shape of the velocity field spatial distribution of turbulence "Tu".

- Please check the meaning?

"Hence the conclusion that by conscious and planned location of plant zones and by appropriately choosing their shape , it is possible to shape the spatial distribution of turbulent kinetic energy (TKE) of the stream in the river. This is of great importance, for example, in the analysis of erosion, transport and sedimentation processes of river debris. Erosion processes of river material are to be expected in this concentrated main stream. On the other hand, in the zones with low intensity of turbulence, sedimentation and accumulation of river debris is possible."

-The part above can be given at the end of the introduction section to highlight the motivation of this research.  

I think, it would be better if the results are clearly and comparatively presented in the conclusion section.
